# Chilling- and dark-regulated photoprotection in *Miscanthus*, an economically important C₄ grass
Jared Haupt [1] & Katarzyna Glowacka [1,2] ✉

Tolerance of chilling dictates the geographical distribution, establishment, and productivity of C₄ crops. Chilling reduces enzyme rate, limiting the sink for the absorbed light energy leading to the need for quick energy dissipation via non-photochemical quenching (NPQ). Here, we characterize NPQ upon chilling in three *Miscanthus* accessions representing diverse chilling tolerance in C₄ grasses. High chilling tolerant accessions accumulate substantial amounts of zeaxanthin during chilling nights in both field and growth chamber settings. Chilling-induced zeaxanthin accumulation in the dark enhances rate of NPQ induction by 66% in the following morning. Based on our data, the emerging ways for the unique regulation of NPQ include post-translational regulation of violaxanthin de-epoxidase (VDE), VDE cofactor accessibility, and absence of transcriptional upregulation of zeaxanthin conversion back to violaxanthin. In the future, more studies will be required to obtain further evidence for these ways contributions to the chilling-dark regulation of NPQ. Engineering dark accumulation of zeaxanthin will help improve crop chilling tolerance and promote sustainable production by allowing early spring planting to maximize the use of early-season soil moisture. Driving the engineered trait by chilling inducible promoter would ensure the minimization of a trade-off between photoprotection and photosynthesis efficiency.

Photosynthesis in C₄ plant species is theoretically the most efficient in terms of water, light, and nitrogen use when fixing $CO_2$[1]. C₄ warm-season grasses have great economic value for food production, with C₄ cereals contributing 42% of the total world cereal production[2]. The species of sugarcane (*Saccharum officinarum*), sorghum (*Sorghum bicolor*), and *Miscanthus* also have potential for bioenergy production[3]. Unfortunately, C₄ grasses typically display poor tolerance to chilling (≥4 °C and ≤15 °C), which limits their geographical distribution, establishment, and productivity[3–5]. Chilling slows enzymatic reactions, causing absorbed energy (light) to impose extensive photodamage through the formation of reactive oxygen species, culminating in plant death[6–8]. Excess absorbed energy can be released harmlessly as heat which can be measured as non-photochemical quenching (NPQ) of chlorophyll fluorescence. NPQ is rapid protection from photodamage[9] that contributes to defense against stress[10,11]. Perennial, rhizomatous grasses from genus *Miscanthus* have a broad natural distribution from the South Pacific to northern China and eastern Russia (up to 50°N)[12,13], making it a useful model to study adaptation to chilling in C₄ grasses. The chilling tolerance of some *Miscanthus* species is exceptional and well-documented[7,14–17], yet how photosynthesis is

protected during the initial response to chilling and whether NPQ has a role are unknown.

While NPQ is prevalent from green algae to land plants, its regulation exhibits stress-inducible variations[11,18,19]. For instance, violaxanthin is converted to zeaxanthin in a light-independent manner in photosynthesizing organisms that are adapted to extremely challenging environments like winter with evergreen trees[10,18], or desiccation and freezing in subalpine resurrection herbaceous plants[20]. However, dark accumulation of zeaxanthin has not previously been considered in economically important C₄ grasses. Here, we demonstrated that accessions of the C₄ grass *Miscanthus* with high chilling tolerance, originating from the northern edge of the distribution of this genus, accumulate zeaxanthin in the dark to increase the speed of NPQ induction in the early morning for better protection from photodamage. Based on our results, multiple possible mechanisms may contribute to the dark accumulation of zeaxanthin, including higher accessibility of the VDE cofactor, lower VDE sensitivity to a reducing environment and absence of transcriptional upregulation of *ZEP* encoding the enzyme converting zeaxanthin back to violaxanthin at night, together with generally higher xanthophyll abundance. Bioengineering of crops with

[1]Department of Biochemistry and Center for Plant Science Innovation, University of Nebraska-Lincoln, Lincoln, NE, USA. [2]Institute of Plant Genetics, Polish Academy of Sciences, 60-479 Poznań, Poland. ✉e-mail: kglowacka2@unl.edu

improved chilling tolerance that could allow early spring planting to increase the use of early-season soil moisture.

## Results and discussion

We investigated NPQ in three *Miscanthus* accessions: *Miscanthus sacchariflorus* 'Robustus-Blumel' (MsaRB), *Miscanthus ×giganteus* 'Illinois' (MxgI) and *Miscanthus sinensis* var. condensatus 'Cosmo Revert' (MsiCR), with high, moderate and low chilling tolerance, respectively. The maximum yield of photosystem II (PSII) efficiency ($F_v/F_m$) decreased significantly after a chilling night in the field in MsaRB but not in MsiCR (Fig. 1a, b; Supplementary Fig. S1). This result was unexpected, as lower $F_v/F_m$ values reflect lower function of PSII reaction centers and would be expected in low-chilling-tolerant accessions[21]. However, $F_v/F_m$ can also decline when other physiological responses compete with charge separation, such as NPQ[11,20]. We determined that the rate of NPQ induction in the light was significantly higher (~44% on average) in accessions with high or moderate chilling tolerance than in the accession with low chilling tolerance after a chilling night (Fig. 1c, d and Supplementary Fig. S2). Under stress conditions the maximal fluorescence ($F_m$) of dark-adapted leaf used in the formula for $F_v/F_m$ [$(F_m-F_o)/F_m$; where $F_o$ is the minimal fluorescence] might be underestimated due to stress-related long-lived quenching[20,22–26]. Our data showed that a decrease in $F_v/F_m$ arose because of induction of NPQ during the chilling-dark treatment, which led to an observed reduction in $F_m$. The difference between MsaRB and MsiCR was even more pronounced in the adjusted NPQ ($NPQ_A$) value (using the values from warm-treated samples to calculate the true value of unquenched fluorescence; see Materials and Methods) that was 133% higher in MsaRB than in MsiCR (Fig. 1c). We confirmed these results in a second independent field trial (Supplementary Fig. S3; Supplementary Fig. S1).

One possible explanation for lower $F_v/F_m$ and faster NPQ induction in response to light is the dark conversion of xanthophyll pigments from their epoxidized form (violaxanthin) to their de-epoxidized forms (antheraxanthin and zeaxanthin), where the zeaxanthin modulates and intensifies the rate of NPQ[20]. In extremely challenging environments the accumulation of de-epoxidized xanthophyll cycle pigments might increase[11], and last overnight[18] or be triggered by a stimulus other than light[19]. To explore this possibility, we performed the xanthophyll pigments quantification via high-performance liquid chromatography on plants subjected to chilling in growth chamber (Fig. 2). We detected significantly more zeaxanthin in leaf extracts of MsaRB than MsiCR at the end of a chilling night but not a warm

night (Fig. 2a). The amount of zeaxanthin rapidly increased after 15 min of light exposure in chilling but remained significantly higher in MsaRB and MxgI than in MsiCR. Exposure to a chilling-light treatment for 15 min after a warm night also induced zeaxanthin formation, but only in MsaRB and to a lesser extent than when the light treatment was preceded by chilling night. Very few zeaxanthin molecules per PSII reaction center are sufficient to compete efficiently for excitation energy with open reaction centers[27]. We confirmed the accumulation of zeaxanthin in response to chilling in additional *Miscanthus* accessions with high or moderate chilling tolerance (MsaRU or MxgN, respectively; Supplementary Fig. S4).

The high zeaxanthin accumulation under chilling in growth-chamber-grown MsaRB was confirmed on field-grown plants using the photochemical reflectance index (PRI), which is negatively correlated with zeaxanthin and antheraxanthin formation and can be measured non-destructively and in a high-throughput manner[28]. MsaRB accumulated significantly more de-epoxidized xanthophyll cycle pigments after a natural chilling night in the field, as estimated from significantly lower PRI values, than did MsiCR (Supplementary Fig. S5; Supplementary Fig. S1). Additionally, we measured significantly higher levels of carotenoids in MsaRB and MxgI (under warm and chilling day conditions both preceded by chilling night; Supplementary Fig. S5) and anthocyanins (warm day preceded by chilling night) in MsaRB, as calculated from hyperspectral reflectance (Supplementary Fig. S5). These results demonstrate that after a chilling night, MsaRB accelerated NPQ induction in the light by accumulating de-epoxidized xanthophyll cycle pigments during the previous night to protect its photosynthetic machinery from photodamage. Upregulation of NPQ via chilling could also contribute to maintaining a balance between source (production of photosynthesis) and sink (consumption of reduced carbon in plant growth)[29]. The increase in the contents of other protective pigments such as carotenoids and anthocyanins may enhance protection against photoinhibition and the photooxidative effects of high light intensity by absorbing high-energy quanta and scavenging free radicals[30,31].

Using samples from the same plants that were subjected to xanthophyll pigment quantification, we determined transcript abundance of the three key NPQ genes (Fig. 3). The *Violaxanthin de-epoxidase* (*VDE*) transcript levels were stable across treatments and accessions, suggesting that regulation of *VDE* expression could not explain the increase in zeaxanthin formation in MsaRB and MxgI (Fig. 3b). *Photosystem II subunit S* (*PsbS*) encodes a protein responsible for energy-dependent quenching, the major

---

**Fig. 1 | Chilling-induced differences in energy quenching among three *Miscanthus* accessions differing in chilling tolerance. a**, **b** Maximum quantum yield of photosystem II ($F_v/F_m$), **c** NPQ induction slope, and **d** NPQ induction curve. In panels c and d, the measured NPQ and adjusted to the initial quenching NPQ ($NPQ_A$) are presented. Leaf discs were incubated overnight at 20 °C or 4 °C, $F_v/F_m$ measured, and exposed to 10 min of light to estimate NPQ induction kinetics. Plants were grown in the field in Urbana-Champaign, IL, USA (40.067 N, 88.198 W). In panel **d**, blue data points are hidden behind red points for MxgI and MsiCR accessions. All numbers represent means ± standard error for *n* = 4 biological replicates. In **b**–**d**, individual biological replicates are represented as circles. Significance was determined by Dunnett's post-hoc test within accessions with warm treatment as control (black) or within treatments with MsiCR as control (corresponding color). *$p < 0.05$; **$p < 0.01$; ***$p < 0.001$. MsaRB, *M. sacchariflorus* 'Robustus-Blumel' (high chilling tolerance); MxgI, *M. ×giganteus* 'Illinois' (moderate tolerance); MsiCR, *M. sinensis* var. condensatus 'Cosmo Revert' (low tolerance). The weather data corresponding to measurements are shown in Supplementary Fig. S1.

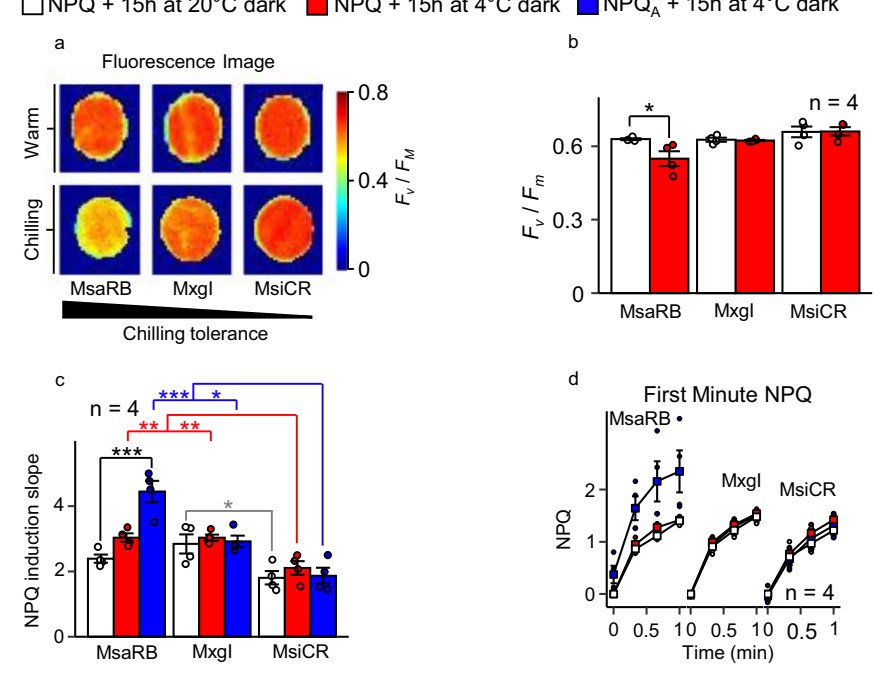

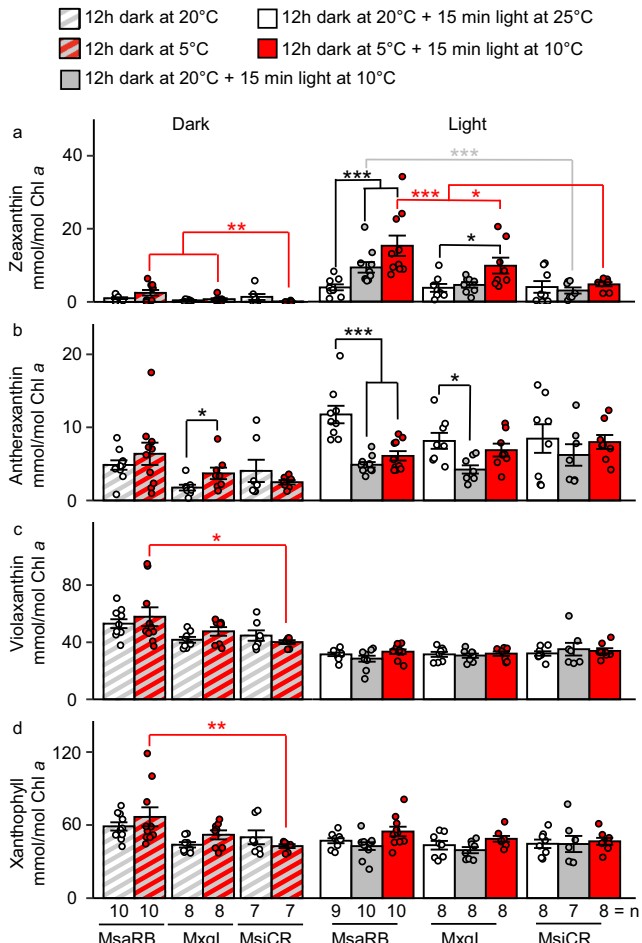

**Fig. 2 | Chilling and light-induced changes in xanthophyll cycle pigment concentrations in three *Miscanthus* accessions differing in chilling tolerance.**
**a** Zeaxanthin; **b** antheraxanthin; **c** violaxanthin and **d** total xanthophyll cycle pigments. Plants grown for 6 weeks at 25 °C/20 °C (day/night) were exposed prior to sampling to one of five combinations of temperature and light treatments: 20 °C in darkness; 5 °C in darkness; 20 °C in darkness +25 °C under 900 µmol m$^{-2}$ s$^{-1}$; 20 °C in darkness +10 °C under 900 µmol m$^{-2}$ s$^{-1}$; 5 °C in darkness +10 °C under 900 µmol m$^{-2}$ s$^{-1}$. Numbers are means ±standard error from 7 to 10 biological replicates. Individual biological replicates are represented as circles. Significance was determined by Conover-Iman test with Benjamin and Hochberg correction across accessions with MsiCR as control (red or gray asterisks) or within each accession with warm treatment as control (black asterisks). *$p < 0.05$; **$p < 0.01$; ***$p < 0.001$. MsaRB, *M. sacchariflorus* 'Robustus-Blumel' (high chilling tolerance); MxgI, *M. ×giganteus* 'Illinois' (moderate chilling tolerance); MsiCR, *M. sinensis* var. condensatus 'Cosmo Revert' (low chilling tolerance).

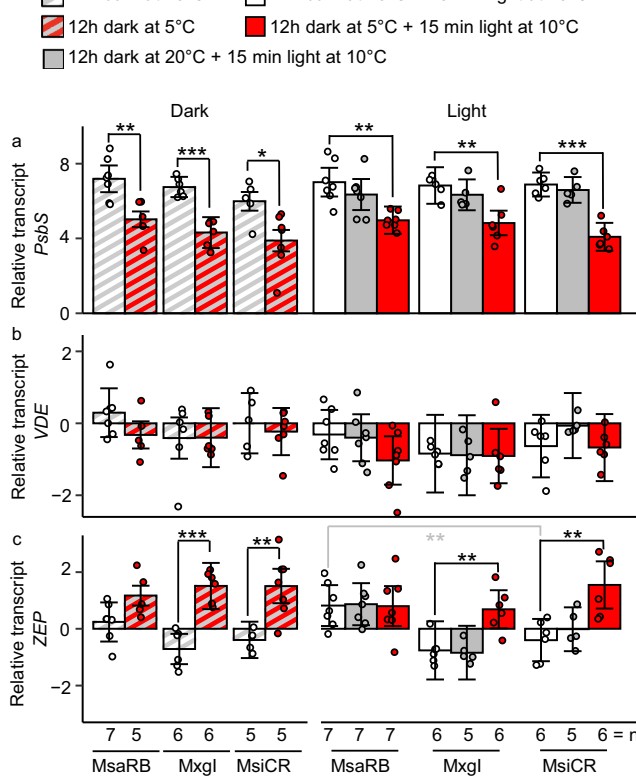

**Fig. 3 | Transcript abundance of the three key NPQ genes in three *Miscanthus* accessions differing in chilling tolerance. a** *Photosystem II subunit S* (*PsbS*), **b** *Violaxanthin de-epoxidase* (*VDE*) and **c** *Zeaxanthin epoxidase* (*ZEP*). Plants grown for 6 weeks at 25 °C/20 °C (day/night) were exposed prior to sampling to one of five combinations of temperature and light treatments: 20 °C in darkness; 5 °C in darkness; 20 °C in darkness +25 °C under 900 µmol m$^{-2}$ s$^{-1}$; 20 °C in darkness +10 °C under 900 µmol m$^{-2}$ s$^{-1}$; 5 °C in darkness +10 °C under 900 µmol m$^{-2}$ s$^{-1}$. Transcript abundance was normalized to the mean between *Ubiquitin* and *Elongation factor 1α* transcript levels. Numbers are means ±standard error from 5 to 7 biological replicates. Individual biological replicates are represented as circles. Significance was determined by a Dunnett's post-hoc test across accessions with MsiCR as control (gray asterisks) or within each accession with warm treatment as control (black asterisks). **$p < 0.01$; ***$p < 0.001$. MsaRB, *M. sacchariflorus* 'Robustus-Blumel' (high chilling tolerance); MxgI, *M. ×giganteus* 'Illinois' (moderate chilling tolerance); MsiCR, *M. sinensis* var. condensatus 'Cosmo Revert' (low chilling tolerance).

and fastest NPQ component[32]. Relative *PsbS* transcript levels decreased in response to a chilling night to a similar degree in all tested accessions, suggesting that energy-dependent quenching did not contribute to the observed increase in MsaRB NPQ rate after chilling night (Fig. 3a). However, relative transcript levels for *Zeaxanthin epoxidase* (*ZEP*), whose encoded enzyme converts zeaxanthin to violaxanthin in the dark, significantly increased following a chilling night compared to a warm night in MxgI and MsiCR but not MsaRB (Fig. 3c). These findings suggest that lower *ZEP* transcript levels (Fig. 3c) and higher xanthophyll pool (Fig. 2d) in MsaRB relative to MsiCR during a chilling night might explain the residual zeaxanthin present in this accession after a chilling night.

As *VDE* transcript levels did not reflect the increased zeaxanthin accumulation or NPQ rate of MsaRB the accessions with high chilling tolerance, we explored the post-translational regulation of VDE using chemical inhibitor assays. For its activation, VDE requires an acidic pH and an oxidizing environment, both conditions promoted by the light reaction

of photosynthesis[33]. An oxidizing environment allows the formation of disulfide bonds that give VDE a more rigid structure and improve its thermal stability and catalysis[34]. We assessed whether *Miscanthus* VDE was active in a reducing environment by infiltrating leaf discs with the reductant dithiothreitol (DTT) before an overnight incubation in chilling or warm conditions prior to fluorescence assays. DTT infiltration lowered the NPQ rates of MsiCR, MxgI and MsaRB under chilling and warm night conditions, although to different extents (Fig. 4). Following DTT treatment and a chilling night, the relative rate of NPQ induction was higher in MsaRB (76%) than MsiCR (23%) (Fig. 4d). The effect of DTT was pronounced under chilling conditions on both the initial slope and the maximum rate of NPQ in MsiCR (Fig. 4a). There were no significant differences between control samples and infiltrated leaf discs with the carotenoid biosynthesis inhibitor norflurazon, regardless of genotype and night temperature (Supplementary Fig. S6a). These results suggest that de novo xanthophyll biosynthesis in response to chilling could not explain the dark accumulation of zeaxanthin; in agreement, the xanthophyll pool did not change significantly in response to chilling in MsaRB or MsaRU relative to the corresponding samples incubated at 20 °C (Fig. 2d and Supplementary Fig. S4). After infiltrating leaf discs with nigericin, a disruptor of the pH gradient across the

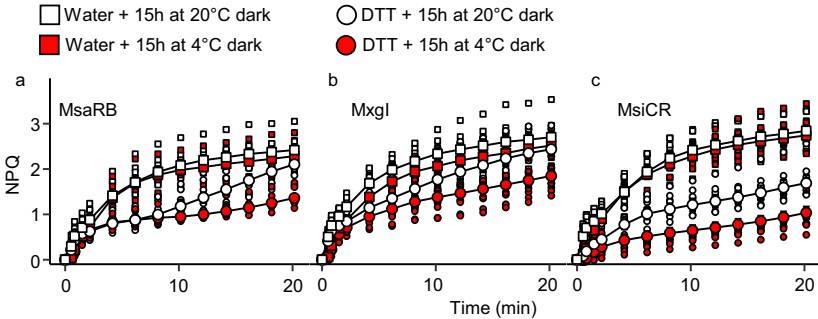

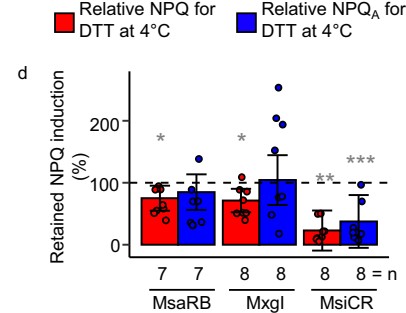

**Fig. 4 | Effect of reductant and chilling night on NPQ the following morning for three *Miscanthus* accessions differing in chilling tolerance.** NPQ induction curve in **a** MsaRB, *M. sacchariflorus* 'Robustus-Blumel' (high chilling tolerance); **b** MxgI, *M. ×giganteus* 'Illinois' (moderate tolerance); **c** MsiCR, *M. sinensis* var. condensatus 'Cosmo Revert' (low tolerance); and **d** NPQ rate in the light estimated from measured NPQ or adjusted to the initial quenching NPQ (NPQ$_A$) and expressed as relative change to the corresponding water treatment. Leaf discs of growth chamber plants were infiltrated with dithiothreitol (DTT) or water (control) and incubated overnight at 4 °C or 20 °C before an NPQ kinetics assay. All numbers represent means ± standard error for n biological replicates = 7–8. In **a–c**, individual biological replicates are represented as small size squares or circles. In **d**, individual biological replicates are represented as circles and a Student's *t* test was performed to compare NPQ induction rate for DTT to water infiltration at 4 °C. $*p < 0.05$; $**p < 0.01$; $***p < 0.001$.

thylakoid membrane, and exposing them to chilling temperature, accessions with high and low chilling tolerance lost similar proportions of the induction rate of NPQ$_A$ (Supplementary Fig. S6b). These results suggest that the pH gradient in MsaRB probably did not persist in darkness. We conclude that MsaRB with high chilling tolerance had faster NPQ rates during chilling mornings because VDE was less sensitive to a reducing environment. While ATPase also possesses the thioredoxin-mediated redox modulation, it cannot be ruled out that the observed higher NPQ in MsaRB after infiltration with a strong reductant is due to chloroplast ATPase being more sensitive to reducing environment which would decline its activity leading to higher light-driven proton motive force, which permits for an increase of NPQ[35,36].

VDE uses the cofactor ascorbate, which is converted to its oxidized form dehydroascorbate during catalysis and then recycled back to ascorbate by dehydroascorbate reductase (DHAR). Although relative *DHAR* transcript levels appeared to be higher in all five treatment combinations for MsaRB than MsiCR, these differences were not significant (Fig. 5a). We also quantified ascorbate (Fig. 5b) and dehydroascorbate (Fig. 5c) in the same samples used for determination of pigments and transcripts. The amount of ascorbate was similar at the end of the chilling night among all three accessions but was significantly higher than at the end of a warm night in MxgI. Plants treated by 15 minutes of chilling in the light in addition to chilling night, similar to those exposed to a chilling night only, had very similar ascorbate contents (Fig. 5b). In contrast to ascorbate, significantly more dehydroascorbate accumulated in MsaRB than MsiCR at the end of a chilling night and after 15 min of exposure to a chilling day after a chilling night (Fig. 5c). MsaRU, like MsaRB, contained significantly more ascorbate at the end of a warm night (~7-fold increase) and significantly more dehydroascorbate at the end of a chilling night (~3-fold increase) than MsiCR (Supplementary Fig. S7). Collectively, our data suggest that an increase in dehydroascorbate, the oxidized form of the VDE cofactor, might indicate higher VDE activity in MsaRB and MsaRU during chilling.

To explore the regulation of NPQ in response to chilling, we examined the transcript levels for the *Miscanthus* orthologs of three recently identified genes associated with NPQ regulation in maize (*Zea mays*)[37]: *Atypical Cys His Rich Thioredoxin* (*ACHT3*), *Phytosulfokine Simulator 3* (*PSI3*) and *Thioredoxin Y1* (*TrxY1*) (Supplementary Fig. S8). *ACHT3* transcript abundance was significantly higher only in one treatment (15 min of light at 4 °C preceded by a warm night) for MsaRB and MxgI the accessions with high and moderate chilling tolerance relative to their corresponding warm controls (Supplementary Fig. S8a). *PSI3* might participate in plant growth in addition to NPQ[38], but its transcript levels did not show significant differences among accessions or treatments except for MxgI (Supplementary Fig. S8b). Therefore, *PSI3* transcriptional regulation probably did not contribute to the NPQ differences in response to chilling reported in this study.

Relative *TrxY1* transcript levels were significantly higher in MsaRB relative to the other accessions in all treatments except chilling night, under which condition the apparent rise did not reach significance (Supplementary Fig. S8c). While the function of TrxY1 is not known, it localizes to the stroma and can promote peroxiredoxin Q (PrxQ) reduction[39]. PrxQ might be involved in sensing the redox state of chloroplasts in *Arabidopsis thaliana*[40]. Collectively, these data suggest that ACHT3 might play a role in the quick adjustment to chilling-light environments, however the exact mechanism it has to be still identified. In principle, TrxY1 and ACHT3 may contribute to redox regulation of VDE in the light.

Based on our results, emerging mechanisms that can contribute to the dark accumulation of zeaxanthin, include higher accessibility of the VDE cofactor, lower VDE sensitivity to a reducing environment and absence of transcriptional upregulation of *ZEP* together with generally higher abundance of xanthophyll cycle pigments (Fig. 6).

Chilling temperatures injure C$_4$ crops in multiple ways, including reduced photosynthetic capacity, leading to necrosis, wilting, and growth inhibition. The future climate is predicted to have on average warmer springs with more frequent severe chilling events. Concurrently, the global population is expected to rise, increasing demand for agricultural products. Crops with improved chilling tolerance would allow for early spring planting, facilitating longer biomass accumulation and maximizing the use of early-season soil moisture. The possible mechanisms emerging here for the unique regulation of NPQ in a C$_4$ grass with high chilling tolerance that is closely related to maize, sorghum and sugarcane can guide the breeding and bioengineering of crops with improved viability and increased resilience to extreme weather events.

## Methods
### Plant materials
Five accessions were chosen from the *Miscanthus* diversity collection of the University of Illinois Urbana-Champaign (UIUC) to represent a broad gradient of chilling tolerance. Two *M. sacchariflorus* accessions, 'Robustus-Blumel' (MsaRB; UI10-00009) and 'RU2012-114' (MsaRU; RU2012-114) show exceptionally high chilling tolerance[7,17]. Two *M. ×giganteus* accessions, 'Illinois' (MxgI; UI10-00107) and 'Nagara' (MxgN; UI10-00123), have moderate chilling tolerance[7,16]. *M. sinensis* var. condensatus 'Cosmo Revert' (MsiCR; UI10-00014) has low chilling tolerance[7]. All rhizomes were multiplied vegetatively in the greenhouse using 6-liter pots (600c, Hummert International, Topeka, KS, USA), filled with growth medium (PRO-MIX, 1038500RG, Premier Horticulture, Quakertown, PA, USA) with the addition of 40 g granulated fertilizer (Osmocote 17-5-11, ICL Growing Solutions, Summerville, SC, USA). In all experiments, measurements and leaf samples were collected from the center third of the youngest fully expanded leaf, avoiding the midrib. Samples for extraction of mRNA, xanthophylls

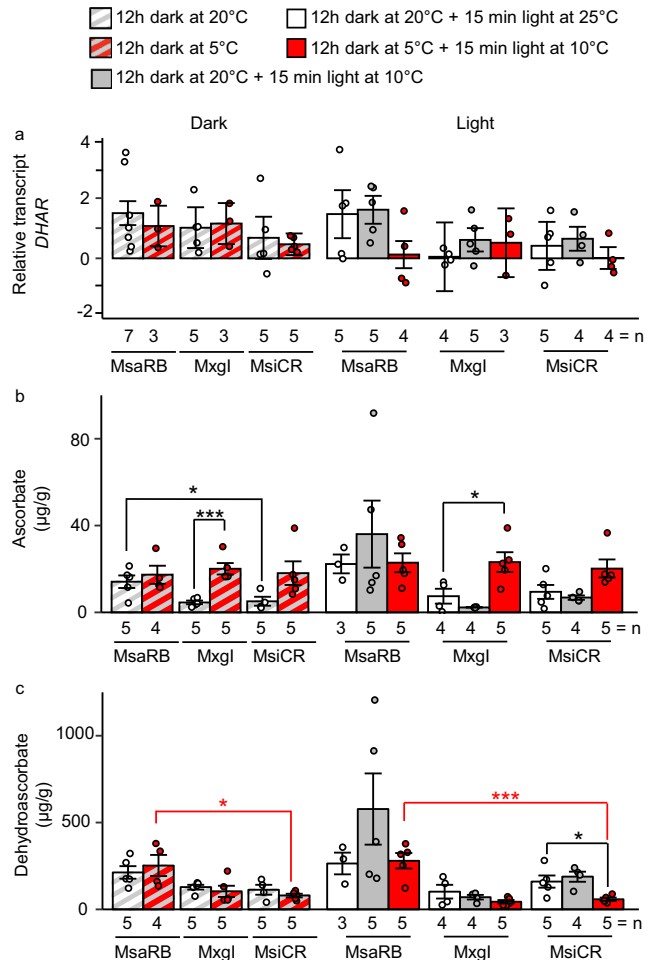

**Fig. 5 | Abundance of molecules related to violaxanthin de-epoxidate (VDE) cofactor in three *Miscanthus* accessions differing in chilling tolerance.** **a** Transcript abundance of *Dehydroascorbate reductase* (*DHAR*); contents of **b** ascorbate and **c** dehydroascorbate. Plants grown for 6 weeks at 25 °C/20 °C (day/night) were exposed prior to sampling to one of five combinations of temperature and light treatments: 20 °C in darkness; 5 °C in darkness; 20 °C in darkness +25 °C under 900 μmol m$^{-2}$ s$^{-1}$; 20 °C in darkness +10 °C under 900 μmol m$^{-2}$ s$^{-1}$; 5 °C in darkness +10 °C under 900 μmol m$^{-2}$ s$^{-1}$. Transcript abundance was normalized to the mean between *Ubiquitin* and *Elongation factor 1α* transcript levels. Numbers are means ± standard error from 3 to 7 biological replicates. Individual biological replicates are represented as circles. Significance was determined by a Dunnett's post-hoc test across accessions with MsiCR as control (red asterisks) or within each accession with warm treatment as control (black asterisks). *$p < 0.05$; ***$p < 0.001$. MsaRB, *M. sacchariflorus* 'Robustus-Blumel' (high chilling tolerance); MxgI, *M. ×giganteus* 'Illinois' (moderate chilling tolerance); MsiCR, *M. sinensis* var. condensatus 'Cosmo Revert' (low chilling tolerance).

and metabolites were snap-frozen in liquid nitrogen and stored at −80 °C until extraction.

## Establishment and management of field trials

The UIUC field experiment (40.067 N, 88.198 W) was planted on drummer silty clay loam on April 29–30, 2015. Single-plant plots were established at a spacing of 2.0 m × 2.0 m in a completely randomized block design with four blocks (2 × 2 blocks) and fertilized each spring with nitrogen fertilizer (80 kg ha$^{-1}$). In each block, MsiCR, MsaRB and MxgI were planted. The collection of leaf discs was performed in summer 2017 from MsaRB, MxgI and MsiCR plants. The field experiment at the University of Nebraska-Lincoln (UNL; 40.829 N, 96.657 W) was planted on silt clay loam soil on June 7, 2019. A completely randomized block design with four blocks was established. Each block contained four single-plant plots of each of the three following

accessions: MsaRB, MxgI and MsiCR. The plots were organized in a 6 × 8 grid with 1.5-m spacing between plots. For both field trials, experimental plots were surrounded by a single row of MxgI; weeds were controlled mechanically and/or with herbicides as needed. Climate data were recorded at weather stations near the field trial locations (Supplementary Fig. S1). The 0.32-cm$^2$ leaf discs used for NPQ kinetics assays were collected with a hole puncher.

## Plant propagation for growth chamber experiments

Rhizomes were divided into 8- to 10-cm pieces and placed in trays with drainage holes (CN-FLXHD-X1, Greenhouse Megastore, Danville, IL, USA) filled using a mix of peat moss:vermiculite:sand:top soil at a ratio of 40:40:15:5). Liquid fertilizer (Peters 20-10-10; 20 general purpose fertilizer, 25#, Peters Inc., Allentown, PA, USA) was applied on the day of transfer of rhizomes fragments into the trays. Trays were watered twice a week or as needed for 2–3 weeks until all accessions sprouted. Plantlets were repotted to tree pots (MT49, Steue and Sons Inc., Tangent, OR, USA) using one of two protocols. For chemical infiltration sampling, plants were potted in growing medium (PRO-MIX) with the addition of 7 g Osmocote 17-5-11 per liter of medium. To induce low soil fertility stress close to that experienced in the UNL field, for sampling for ascorbate/dehydroascorbate, mRNA and xanthophyll content, plants were grown on growth medium (PRO-MIX) without the addition of fertilizer. Pots were randomized and placed in a growth chamber (PGC20, Conviron, Manitoba, Canada) with day/night cycles: 25 °C/20 °C, 900/0 μmol m$^{-2}$ s$^{-1}$, 65% humidity, and a day length of 15 h (chemical infiltration) or 12 h (ascorbate/dehydroascorbate, mRNA and xanthophyll quantification). Plants were grown for an additional 2 weeks, with watering and shuffling of pots every 2 days. Plants were then subjected to one of three air temperature treatments: 25 °C/20 °C day/night (warm control), 10 °C/20 °C (chilling day/warm night) or 10 °C/5 °C (chilling day and night). Samples were collected from plants in the dark and in the light. Treatments were initiated at beginning of night. Dark samples were collected 30 min before the end of the dark period. Light samples were collected 15 min after switching on the lights. Ascorbate/dehydroascorbate, mRNA and xanthophyll sampling were performed for all treatments. For chemical infiltration, leaves were collected from plants grown in the warm control condition and incubated at 4 °C by placing them in a refrigerator.

## NPQ kinetics on leaf discs

As previously described[37], NPQ kinetics were investigated on leaf discs collected into 96-well plates (781611; BrandTech Scientific, Essex, CT, USA). Leaf discs were positioned with their adaxial surface down, covered with moist sponges and incubated overnight in the dark at 21 °C (control) or 4 °C (chilling). Plates with leaf discs were imaged the following day using a modulated chlorophyll fluorescence imager (FluorCam FC 800-C, Photon Systems Instruments, Drasov, Czech Republic; or with a CF Imager, Technologica, Colchester, UK, for UIUC field samples). Chilling-treated plates were allowed to equilibrate to room temperature for 20 min to prevent reflection from condensation. First, the minimum ($F_o$) and maximal ($F_m$) fluorescence in the dark were measured. Subsequently, leaf discs were exposed to 2000 μmol m$^{-2}$ s$^{-1}$ for 10 min, followed by 10 min in darkness. Saturating flashes of 3200 μmol m$^{-2}$ s$^{-1}$ (FluorCam FC 800-C provided by a cool white 6500 K light) or 6000 μmol m$^{-2}$ s$^{-1}$ (CF Imager; provided by light $\lambda_{max}$ = 470 nm) were used in the light and dark periods to capture changes in steady-state ($F_s$) and maximum ($F_m'$) fluorescence over time. Saturating flashes were provided at the following intervals (in min after light exposure began) for CF Imager: 0, 0.33, 0.67, 1, 2, 3, 4, 5, 6, 7, 8, 9, 10, 10.33, 10.67, 11, 12, 13, 16, 19, 22; for FluorCam FC 800-C: 0, 0.263, 0.413, 0.747, 1.08, 2.08, 3.08, 4.08, 5.08, 6.08, 7.08, 8.08, 9.08, 10.08, 10.263, 10.413, 10.913, 11.913, 15.08, 20.08.

NPQ was calculated using Eq. (1), assuming the Stern–Volmer quenching model:

$$NPQ = \frac{F_m}{F_m'} - 1 \qquad (1)$$

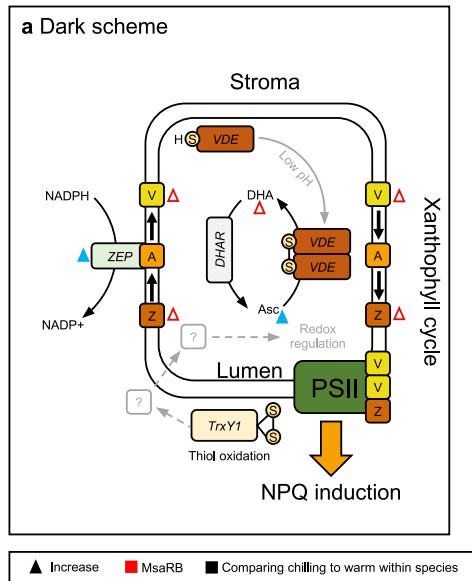
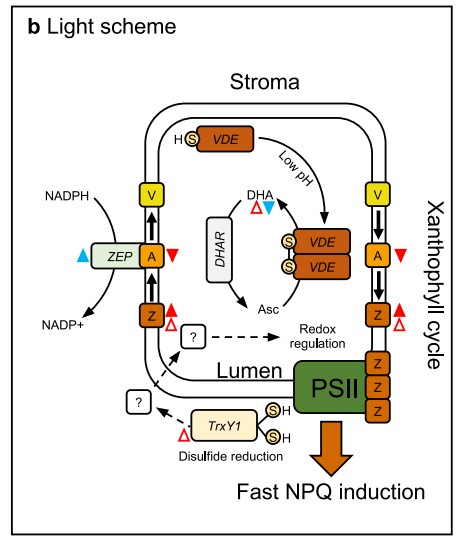

**Fig. 6 | A model showing mechanisms that can contribute to NPQ regulation in chilling-dark and chilling-light conditions in two *Miscanthus* accessions differing in chilling tolerance. a** During overnight chilling exposure, MsaRB maintains residual violaxanthin de-epoxidate (VDE) activity due to lower sensitivity to a reducing environment, leading to higher accumulation of the oxidized form, dehydroascorbate, of the VDE cofactor and higher accumulation of zeaxanthin (Z) relative to MsiCR. In addition, zeaxanthin epoxidase (ZEP) is constitutively active and converts quenching zeaxanthin back to violaxanthin (V) via the intermediate antheraxanthin (A); *ZEP* is transcriptionally upregulated in MsiCR, but not in MsaRB, which displays a higher xanthophyll pull during the chilling night that contributes to the partial retention of zeaxanthin seen in MsaRB at the end of the chilling night. When zeaxanthin binds to antenna complexes, excitation energy is dissipated, $F_v/F_m$ decreases and PSII is protected from photodamage more quickly during light exposure the following morning. **b** Upon 15 min of light exposure, as in dark-chilling conditions, VDE activity is higher in MsaRB than in MsiCR, as supported by the higher accumulation of DHA and zeaxanthin in MsaRB relative to MsiCR. The increase in zeaxanthin in the dark to light transition can be aided by redox regulation, in which *Thioredoxin Y1* (*TrxY1*) might be involved. *TrxY1* is transcriptionally upregulated in MsaRB relative to MsiCR. Changes in metabolite and transcript levels are shown relative to control treatment in the same accession (filled symbols) or within the chilling treatment relative to negative control of MsiCR (empty symbol). Triangles pointing up indicate increased abundance; triangles pointing down show decreased abundance. MsaRB is shown in red symbols, and MsiCR is shown in blue. DHAR, Dehydroascorbate reductase. MsaRB, *M. sacchariflorus* 'Robustus-Blumel' (high chilling tolerance); MxgI, *M. ×giganteus* 'Illinois' (moderate chilling tolerance); MsiCR, *M. sinensis* var. condensatus 'Cosmo Revert' (low chilling tolerance).

The maximum PSII efficiency ($F_v/F_m$) was estimated using the following equation:

$$F_v/F_m = (F_m - F_o)/F_m \qquad (2)$$

NPQ values adjusted to initial quenching, referred to here as $NPQ_A$, were calculated using Eqs. (3)–(5).

$$NPQ_A \text{ at dark point} = \frac{F_{o\,cold}\left(\frac{F_{v\,warm}}{F_{m\,warm}}\right) / \left(1 - \frac{F_{o\,cold}}{F_{m\,cold}}\right)}{F_{o\,cold}\left(1 - \left(\frac{F_{v\,warm}}{F_{m\,warm}}\right)\right)} - 1 \qquad (3)$$

$$NPQ_A \text{ at light points} = \frac{F_{m\,true}}{F_{m}'} - 1 \qquad (4)$$

Where $F_{m\_true}$ was calculated according to Eq. (5)

$$F_{m\,true} = F_{m\,cold}\left(NPQ_A \text{ at dark point} + 1\right) \qquad (5)$$

The NPQ and $NPQ_A$ curves were fit to a hyperbolic equation[37] using MATLAB (Matlab R2019b; MathWorks, Natick, MA, USA) to obtain parameter attributed to rate of kinetics.

### Chemical infiltration
Leaf discs from plants grown in a growth chamber under 25 °C/20 °C day/night cycles were collected using a hole puncher and placed into a 25-ml syringe containing 10 ml one of three solutions: dithiothreitol (DTT; 5 mM; 43815, Sigma, St. Louis, MO, USA), norflurazon (5 µM; 34364, Sigma) or nigericin (100 µM; N7143, Sigma). DTT was dissolved in water, while nigericin and norflurazon were dissolved in 5% (v/v) ethanol with 10 mM KNO₃. The corresponding solvent for each chemical was used as a control treatment. A vacuum was repeatedly pulled and released until the discs sank. The NPQ kinetics of leaf discs were measured and analyzed as described above.

### Leaf reflectance indices
Leaf reflectance was measured on field-grown plants with a leaf spectrometer (CI-710S SpectraVue; CID BioScience Inc., Camas, WA, USA) according to the manufacturer's protocol. After instrument calibration, reflectance values were recorded within 5 s of clamping the leaf into the cuvette to minimize NPQ relaxation. Sub-nanometer reflectance values were converted to integers and averaged before calculating parameters. The following parameters were calculated from the reflectance (R) values: photochemical reflectance index (PRI) = ($R_{531nm}$ − $R_{570nm}$) / ($R_{531nm}$ + $R_{570nm}$), anthocyanin reflectance index (ARI) = ($R_{800nm}/R_{550nm}$ − 1/$R_{700nm}$) and carotenoid reflectance index (CRI) = ($1/R_{510nm}$) − ($1/R_{550nm}$). The light intensity corresponding to the measured reflectance was recorded with a MultispeQ v2.0 instrument (PHOTOSYNQ INC., East Lansing, MI, USA). Both instruments were within 15 cm of one another and leaf angle was maintained.

### Xanthophyll quantification
Tissue was weighed in a 2-ml tube, extracted with 500 µl 100% acetone (A9491, Fisher) and homogenized for 10 min at 20 Hz using a TissueLyser II (Qiagen). Extracted tissue was centrifuged at $16,000 \times g$ (centrifuge 5424, Eppendorf, Hamburg, Germany) for 5 min at 4 °C, and the supernatant was

transferred to an HPLC vial. Extraction of the pellet was repeated, and the supernatants were pooled. Samples were dried under nitrogen stream in a Microvap Microplate Evaporator (11801, Organomation, Berlin, MA, USA). Liquid chromatography separation was performed on a Spherisorb S5 ODS1 (4.6 mm × 250 mm, 5 μm, PSS830615, Waters, Milford, MA, USA) flowing at 1.2 ml/min at room temperature using a quaternary gradient on a UPLC Infinity II (Agilent, Santa Clara, CA) equipped with a diode array detector (DAD). The gradient of the mobile phases (A) 0.1 M Tris base, pH 8.0 (T1503, Sigma), (B) 100% acetonitrile (A955, Fisher), (C) 100% methanol (A456, Fisher) and (D) 100% ethyl acetate (E195, Fisher) was as follows: 14% A, 84% B, 2% C, 0% D to 0% A, 15% B, 60% C, 25% D (all v/v) over 15 min, then held for 3 min before returning to initial conditions for 1 min. Carotenoids were detected using DAD at 440 nm. For quantification, an external standard curve was prepared using a series of standard samples containing different concentrations of carotenoid mixtures. The lower limit of quantification and detection for each compound are as follow: 21 μg/ml for lutein (10010811, Cayman Chemical, Ann Arbor, MI, USA), antheraxanthin (PPS-ANTH, DHI, Portland, OR, USA) and zeaxanthin (10009992, Cayman Chemical) and 20 μg/ml for violaxanthin (PPS-VIOL, DHI). Chlorophyll $a$ and $b$ (PPS-CHLA and PPS-CHLB, respectively, DHI) were measured to normalize experimental variation, with lower limits of quantification and detection of 35 μg/ml and 25 μg/ml for chlorophyll $a$ and $b$, respectively.

### Gene expression

Leaf tissue was ground to a fine powder before total RNA was extracted using a NucleoSpin RNA/Protein kit (740933, MACHEREY-NAGEL GmbH & Co., Duren, Germany) followed by TURBO DNA-free kit treatment (AM1907, Thermo Fisher, Waltham, MA, USA) to remove contamination by genomic DNA. First-strand cDNA was synthesized with oligo(dT) primers using a SuperScript III First-Strand Synthesis kit (18080051, Thermo Fisher). Gene expression was performed by RT-qPCR (1855201, BioRad CFX Connect Real-Time PCR Detection System, BioRad, BioRad, Hercules, CA) using primers designed to target gene sequences conserved across the grasses including the *Miscanthus* genome (*M. sinensis* doubled haploid DH1; IGR-2011-001') and ten transcriptomes from eight related grass species: *Chasmanthium laxum, Panicum hallii, Panicum virgatum, Sorghum bicolor, Sorghum bicolor Rio, Sorghum bicolor RTx430, Setaria italica, Setaria viridis, Urochloa fusca,* and *Zea mays* (Joint Genome Institute). Primer sequences are given in Supplementary Table S1. The *Ubiquitin* and *Elongation factor-1-α* (*Ef1a*) genes were used for normalization in ΔΔCt method.

### Ascorbate and dehydroascorbate contents

Tissue was pulverized in a TissueLyser II ball mill (85300, Qiagen, Germantown, MD, USA) at 20 Hz using a 30-s cycle (1 s shaking, 1 s still, repeated 15×). Pulverized tissue was transferred to 2-ml Eppendorf tubes and stored at −80 °C. Tissue weight was determined by adding powder directly to cell disruption buffer, containing 2.1 μM $^{13}C5$-$^{15}N$-proline (CLM-1396, Cambridge Isotope Laboratories, Tewksbury, Massachusetts, USA), 12 μM $^{13}C6$-glucose (110187-42-3, Sigma) as internal standards for LC-MS, plus 50–100 ng 0.5-mm ZrO beads (ZROB05, Next Advance, Troy, New York, USA). Samples were processed at 4 °C in groups of 24 using a bullet blender (BB24AU, Next Advance) with three 3-min cycles on setting "9". Samples were centrifuged at $10,000 \times g$ for 10 min at 4 °C (centrifuge EP-5415D, Eppendorf, Hamburg, Germany). Supernatants were transferred into Eppendorf tubes and concentrated by evaporation on a RotoVap (Fisher Scientific) for 8 h. The pellets were dissolved into 50 μl LC-grade water (W4502, Fisher Scientific), vortexed for 30 s, and transferred into 360-μl polypropylene V-vials for LC-MS/MS analysis (03452358, Fisher). Capped vials were placed into the autosampler of an LC-1200 instrument (Agilent Technologies, Santa Clara, CA) coupled to an Amide XBridge XP column (2.7 μm, 100 × 4.6 mm) for separation and analysis, in turn coupled to a 4000 QTrap (Sciex, Framingham, MA, USA) operating in MRM mode with both positive and negative ionization modes. The mobile phase

consisted of (A) LC-grade acetonitrile and (B) 20 mM ammonium acetate, 20 mM ammonium hydroxide pH 9.5 in LC-grade water (A955, A669S, W6500 respectively, Fisher). Data were imported into Multiquant 3.0 (Sciex) to extract the peak areas used for quantification and normalization. Quantification was carried out with external calibration curves using the compounds of interest while the normalization was done using the sample weights in the final 50-μl extracts.

### Statistics and reproducibility

Normality and homoscedasticity were verified with the Shapiro–Wilk and Brown–Forsythe tests, respectively. If the null hypothesis was discarded by either test, data were transformed or the Wilcoxon nonparametric test was applied. Significant effects as determined by two-way or three-way analysis of variance (ANOVA, $\alpha = 0.05$) were followed by testing accession means against the accession with low chilling tolerance i.e. MsiCR (inter-accessions; $\alpha = 0.05$) or against the warm temperature condition (intra-accessions; $\alpha = 0.05$) using Dunnett's test with a multiple comparison correction for normally distributed data and a Wilcoxon nonparametric test for non-normally distributed data. Statistical analyses were performed in R version 4.1.2 using the package DescTools 0.99.45 and PMCMRplus 1.9.6. All numeric data are presented as means ± standard error for $n$ = from 3 to 10 biological replicates.

### Reporting summary

Further information on research design is available in the Nature Portfolio Reporting Summary linked to this article.

## Data availability

Raw data supporting the findings of this manuscript are available in the supplementary information files as Supplementary Data 1–12. All other data are available from the corresponding author on reasonable request. A reporting summary for this Article is available as Supplementary Material.

## Code availability

Matlab code to fit NPQ induction to hyperbolic equation is available in Supplementary Information file as Note 1 and in public depository[41]. The version R2019b of Matlab was used to generate the outcome.

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

## Acknowledgements

The authors thank Eric Sacks for access to *Miscanthus* accessions; Johannes Kromdijk for fruitful discussion and guidance in the calculation of $NPQ_A$; Matt Anderson, Alexander Batelaan, Bailey McLean and Annie Nelson for assistance during field and greenhouse measurements; Nebraska Center for Biotechnology led by Sophie Alvarez for the HPLC analysis; and TJ McAndrew for assistance in maintaining the field experiment. We thank Javier Seravalli at the UNL Redox Biology Center for quantification of ascorbate and dehydroascorbate. This project was supported by the National Science Foundation under awards OIA-1557417 (EScoR FIRST) and IOS-2142993 (CAREER).

## Author contributions

K.G. conceived the project. J.H. and K.G. collected the data. J.H. analyzed the data and designed figures. J.H. and K.G. drafted the manuscript. K.G. obtained funding for the project.

## Competing interests

The authors declare no competing interests.
