## [Transparent Peer Review file · Communications Biology]

Chilling- and dark-regulated photoprotection in Miscanthus, an economically important C4 grass

Corresponding Author: Dr Katarzyna Glowacka

Version 0:

Reviewer comments:

Reviewer #1

(Remarks to the Author)

The manuscript entitled “Chilling- and dark-regulated photoprotection in an economically important C4 grass” found that NPQ played a critical role in chilling tolerance, which provides a new method for increasing cold tolerance for plants. My main concerns:

1. All of the results in the manuscript are physiological data, these results lack genetic evidence. As shown in the abstract “The possible mechanisms uncovered here for the unique regulation of NPQ....”. It is difficult to draw reliable conclusions without genetic verification.
2. The Fv/Fm value in the cold tolerant variety was the lowest, the reasons should be discussed in detail.
3. Materials and methods: The authors just describe the materials and methods used in this article, others are not used and should be deleted, such as F2 progenies.

Reviewer #2

(Remarks to the Author)

The manuscript “Chilling- and dark-regulated photoprotection in an economically important C4 grass” demonstrated that chilling tolerant *Miscanthus* accessions induce a mechanism in cold and darkness that quenches fluorescence both when measured directly from darkness (Fv/Fm) and enhances the fluorescence quenching measured in light (NPQ). This is shown to be associated with zeaxanthin accumulation via a previously uncharacterized mechanism. The authors attempted to elucidate the mechanism by manipulating the redox state with DTT and by examining the expression of NPQ-related genes by qPCR.

The observation of a dark-inducible, probably zeaxanthin-related, dissociation mechanism is interesting and important. However, the manuscript as it stands is very difficult to read. There are too many panels in the figures and the figures are very difficult to understand based on the captions. The division of the data into different figures is also illogical and, for example, Figure 1A shows the same thing several times without any clear added value. Moreover, HPLC is a well-established way of measuring carotenoids and would fit before the hyperspectral image and hyperspectral can be a supplement, but it adds no value to the story itself over the HPLC data etc.

The story should be more logical and the images easier to read. I don't think it's the job of the reviewer to give detailed advice on rewriting.

Minor comment: Row 35 and 36: The dissipation of excess excitation energy is measured as non-photochemical quenching of fluorescence. NPQ is a way of measuring heat dissipation indirectly from fluorescence quenching, not the mechanism for dissipating energy as heat.

Reviewer #3

(Remarks to the Author)

This manuscript by Haupt and Glowacka reports on observations that the higher chilling tolerance of some *Miscanthus* accessions is associated with higher non-photochemical quenching (NPQ). The chilling tolerant accessions accumulated

higher amounts of zeaxanthin during chilling nights that is proposed to offer photoprotection the following morning. The authors characterize explore a number of possible mechanisms that could contribute to increased zeaxanthin accumulation, with the collective evidence favoring a reduced rate of conversion of zeaxanthin to violaxanthin. Slower turnover and dark accumulation of zeaxanthin has been proposed to contribute to photosynthetic activity in other winter-active plants, but this report is the first demonstration it occurs in a C4 grass.

Overall, the manuscript is clearly written and the data presentation in both the main and supplemental figures is easy to follow. The main conclusions of the work, that the chilling-tolerant MsaRB and MsaRU accessions exhibit higher NPQ and accumulate more zeaxanthin than chilling-sensitive Mxg and MsiCR, are strongly supported. The authors then go on to measure other metabolites and transcripts encoding genes associated with the zeaxanthin pathway, as well as NPQ responses to inhibitors of zeaxanthin synthesis (norflurazon) or thylakoid function (nigericin). Collectively, the results point to a higher activity of violaxanthin de-epoxidase enzyme activity in the chilling-tolerant accessions due to reduced sensitivity to inhibition by a reducing environment. Although a reasonable hypothesis, the evidence for this mechanism is currently indirect.

The significance of this work is that it offers another explanation for the remarkable chilling tolerance of C4 photosynthesis in some *Miscanthus* accessions, a trait that is often proposed to be a major contributor to the adaptation of *Miscanthus* and its high biomass yields in cultivation. The vast majority of prior studies of chilling tolerant photosynthesis in *Miscanthus* have focused on greater thermostability and activity of the C4 carboxylation enzymes (PPDK and/or Rubisco), but direct genetic evidence in support of this mechanism also still remains elusive. Although additional experiments may be needed to better define the strategy, the information shared here can inform further studies that exploit natural variation or genetic engineering strategies to enhance the chilling tolerance of related C4 grass crops such as sugarcane, sorghum, or maize.

The manuscript would benefit from clarification of the following issues or questions:

1. In Figure 1a, are there values for NPQA in plots for Mxg1 and MsiCR in the bottom panels that are overlapping with the other two treatments and thus not readily visible? That appears to be implied by the plot of NPQ induction slope in the upper right part of the figure. If the blue data points are just obscured, then perhaps the legend should indicate that, or replot so that the NPQA values are more visually apparent to the reader.
2. For the data obtained from the field experiments in Figure S2, details about the date, time of day, plant growth stage and weather conditions surrounding the time of sampling were not provided. However, context for this experiment appears in Figure S7. The authors should introduce Figure S7 earlier and as it is currently not mentioned anywhere in the main text, should be called out in the Material and Methods, probably at Line 71.
3. For Figure S6, the order of the panels is not the same as described in the figure legend and the associated text in Lines 145-159. It appears panels a and c need to be swapped.
4. One question that arises for the RT-PCR assays in *Miscanthus*, where the studied accessions vary in both ploidy (Msa tetraploid, Mxg triploid, Msi diploid) and possibly alleles, is whether all possible copies of the gene are amplified by the assay primers in all accessions? The description of RT-PCR primer design in lines 460-464 is confusing on this point, where it is stated that degenerate primers were designed to target shared sequences across multiple grass species, yet Table S1 lists only one sequence for each primer without any indication of degenerate bases. Did the authors actually mean the opposite, that primers were designed to target gene sequences conserved across the grasses? If degenerate primers were used, then the positions of the degenerate bases should be indicated in Table S1.

Version 1:

Reviewer comments:

Reviewer #1

(Remarks to the Author)

I am satisfied with the authors' response and the manuscript can be accepted for publication.

Reviewer #2

(Remarks to the Author)

The manuscript has been revised according to the reviewers' comments, has been significantly improved and is now a nice, compact story. The manuscript presents a new phenomenon that is physiologically interesting and potentially very important for future applications. Overall, the data and the interpretation of the data are rather superficial, but on the other hand the main observation is important.

Here are some minor comments.

Line 20: I would soften the statement in the last sentence of the abstract. There is always a trade-off between protection and efficiency.

Line 34: It is still debatable how sensitive NPQ mutants (PSBS and xanthophyll cycle) are and how well reaction centre quenching can compensate for the lack of these classical NPQ mechanisms.

Results and discussion: Traditionally, NPQ mechanisms have been seen as protective mechanisms for light reactions, but as this manuscript and some other articles suggest, NPQ may be more important in maintaining source-sink balance,

preventing adverse stromal responses and being involved in the control of many physiological responses in which chloroplasts are involved. It would be good to have a little more discussion about these questions whether NPQ is a photoprotective mechanism of light responses or whether it has a more important role in source-sink regulation and in chloroplast-related physiology.

Both redox reagents and nigericin were used. However, the redox regulation of ATP synthase was not discussed. How NPQ is induced upon light onset is strongly dependent on the state of ATP synthase and this should be taken in account in the interpretation of results and in discussion.

Reviewer #3

(Remarks to the Author)

The revised manuscript is much improved in its clarity, particularly Figure 1. The authors satisfactorily addressed the comments raised in my first review. I do offer the following suggested minor edits:

1. The proposed change to the abstract to soften the conclusion of “mechanisms” for regulation of NPQ are beneficial, but perhaps could be worded more clearly. I suggest, “Further studies obtained evidence for post-translational modifications to VDE, VDE cofactor accessibility, and lack of upregulation of zeaxanthin epoxidase as possibly contributing to this unique regulation of NPQ.”
2. The new text in first paragraph of Results should revise verb tenses, “use” should be “used”, “long-living” should be “long-lived” or “prolonged quenching induced by stress”.
3. Bottom paragraph of page 4 Results: “Transcript of VDE transcript levels”, “VDE transcript levels”

Version 2:

Reviewer comments:

Reviewer #2

(Remarks to the Author)

The authors have satisfactorily taken my comments into account and revised the manuscript accordingly.

Point-by-point response to the referees' comments

Referee's original text in black
Our responses in blue

Reviewers' comments:

Reviewer #1 (Remarks to the Author):

The manuscript entitled "Chilling- and dark-regulated photoprotection in an economically important C4 grass" found that NPQ played a critical in chilling tolerance, which provides a new method for increasing cold tolerance for plants. My main concerns:

1. All of the results in the manuscript are physiological data, these results lack genetic evidence. As shown in the abstract "The possible mechanisms uncovered here for the unique regulation of NPQ...". It is difficult to draw reliable conclusions without genetic verification.

We agree with the reviewer that our conclusions might sound too strong compared to the evidence that we presented. Therefore, in the revised version of the manuscript, we now say that our physiological, molecular (xanthophylls, mRNA and metabolite quantifications) and biochemical (inhibitory assays) results suggest three possible ways of the unique regulation of NPQ instead of saying that we "uncovered unique mechanisms of regulation". Accordingly, the text has been reworded in the lines L19-20; 179-182; L188-L191 and the lines L43-45 in the previous version of manuscript have been deleted.

Unfortunately, there is no publicly available genomic information for *M. sinensis* and *M. sacchariflorus* accessions investigated in our paper which we could use to add genetic verification. Furthermore, *Miscanthus* is an ancestral tetraploid that complicates genomic sequence analyses. Therefore, due to time constrain of the revision of this paper we rather made our conclusions more suggestive than sequence the genome of *Miscanthus sacchariflorus* Robustus-Blumel (MsaRB), and *Miscanthus sinensis* var. *condensatus* 'Cosmo Revert' (MsiCR).

2. The F_v/F_m value in the cold tolerant variety was the lowest, the reasons should be discussed detailed.

Additional text has been added to the Results and Discussion section to discuss the reason for the F_v/F_m decline.

"Under stress conditions the maximal fluorescence (F_m) of dark-adapted leaf use in the formula for F_v/F_m [$(F_m - F_o)/F_m$; where F_o is the minimal fluorescence] might be underestimated due to stress-related long-living quenching^{20,22-26}. Our data showed that a decrease in F_v/F_m arose because of induction of NPQ during the chilling-dark treatment, which led to an observed reduction in F_m ."

3. Materials and methods: The authors just describe the materials and methods used in this article, others are not used should be deleted, such as F2 progenies.

Thank you for this comment. The information about F2 progenies has been removed from the revised version of the manuscript.

Reviewer #2 (Remarks to the Author):

The manuscript “Chilling- and dark-regulated photoprotection in an economically important C4 grass” demonstrated that Chilling tolerant *Miscanthus* accessions induce a mechanism in cold and darkness that quench fluorescence both when measured directly from darkness (F_v/F_m) and enhances the fluorescence quenching measured in light (NPQ). This is shown to be associated with Zeaxanthin accumulation via previously uncharacterized mechanism. The authors attempted to elucidate the mechanism by manipulating the redox state with DTT and by examining the expression of NPQ-related genes by qPCR.

The observation of a dark-inducible, probably zeaxanthin-related, dissociation mechanism is interesting and important. However, the manuscript as it stands is very difficult to read. There are too many panels in the figures and the figures are very difficult to understand based on the captions. The division of the data into different figures is also illogical and, for example, Figure 1A shows the same thing several times without any clear added value. Moreover, HPLC is a well-established way of measuring carotenes and would fit before the hyperspectral image and hyperspectral can be a supplement, but it adds no value to the story itself over the HPLC data etc.

The story should be more logical and the images easier to read. I don't think it's the job of the reviewer to give detailed advice on rewriting.

Thank you for the comment. We agree with the reviewer that the complexity of the figures might have compromised their clarity. Therefore, we have removed panel A of Fig. 1 the full induction curves and kept the zoom-in on the initial portion of NPQ curves showing the rate of the induction that is discussed in the results.

According to the reviewer's suggestion, we have moved hyperspectral data results to Supplementary Material. In the Results section, we also placed the results from HPLC before the hyperspectral data. The HPLC data became its own figure.

In addition, we split Fig. 2 into two separate figures. The panel c of Fig. 1 became its own figure. We believe that collectively, these changes made the corresponding captions easier to follow and figures easier to read.

Minor comment: Row 35 and 36: The dissipation of excess excitation energy is measured as non-photochemical quenching of fluorescence. NPQ is a way of measuring heat dissipation indirectly from fluorescence quenching, not the mechanism for dissipating energy as heat.

Thank you for pointing out this miswording which has been corrected in the revised version of manuscript. The new text is as follows: “Excess absorbed energy can be released harmlessly as heat which can be measured as non-photochemical quenching (NPQ) of chlorophyll fluorescence.”

Reviewer #3 (Remarks to the Author):

This manuscript by Haupt and Glowacka reports on observations that the higher chilling tolerance of some *Miscanthus* accessions is associated with higher non-photochemical quenching (NPQ). The chilling tolerant accessions accumulated higher amounts of zeaxanthin during chilling nights that is proposed to offer photoprotection the following morning. The authors characterize explore a number of possible mechanisms that could contribute to increased zeaxanthin accumulation, with the collective evidence favoring a reduced rate of

conversion of zeaxanthin to violaxanthin. Slower turnover and dark accumulation of zeaxanthin has been proposed to contribute to photosynthetic activity in other winter-active plants, but this report is the first demonstration it occurs in a C4 grass.

Overall, the manuscript is clearly written and the data presentation in both the main and supplemental figures is easy to follow. The main conclusions of the work, that the chilling-tolerant MsaRB and MsaRU accessions exhibit higher NPQ and accumulate more zeaxanthin than chilling-sensitive Mxg and MsiCR, are strongly supported. The authors then go on to measure other metabolites and transcripts encoding genes associated with the zeaxanthin pathway, as well as NPQ responses to inhibitors of zeaxanthin synthesis (norflurazon) or thylakoid function (nigericin). Collectively, the results point to a higher activity of violaxanthin de-epoxidase enzyme activity in the chilling-tolerant accessions due to reduced sensitivity to inhibition by a reducing environment. Although a reasonable hypothesis, the evidence for this mechanism is currently indirect.

The significance of this work is that it offers another explanation for the remarkable chilling tolerance of C4 photosynthesis in some *Miscanthus* accessions, a trait that is often proposed to be a major contributor to the adaptation of *Miscanthus* and its high biomass yields in cultivation. The vast majority of prior studies of chilling tolerant photosynthesis in *Miscanthus* have focused on greater thermostability and activity of the C4 carboxylation enzymes (PPDK and/or Rubisco), but direct genetic evidence in support of this mechanism also still remains elusive. Although additional experiments may be needed to better define the strategy, the information shared here can inform further studies that exploit natural variation or genetic engineering strategies to enhance the chilling tolerance of related C4 grass crops such as sugarcane, sorghum, or maize.

Thank you for a very positive assessment of our work.

The manuscript would benefit from clarification of the following issues or questions:

1. In Figure 1a, are there values for NPQA in plots for MxgI and MsiCR in the bottom panels that are overlapping with the other two treatments and thus not readily visible? That appears to be implied by the plot of NPQ induction slope in the upper right part of the figure. If the blue data points are just obscured, then perhaps the legend should indicate that, or replot so that the NPQA values are more visually apparent to the reader.

Thank you for pointing out this issue. We have added the text to the legend of Fig. 1 explaining that the blue data points are hidden behind red points for MxgI and MsiCR accessions.

2. For the data obtained from the field experiments in Figure S2, details about the date, time of day, plant growth stage and weather conditions surrounding the time of sampling were not provided. However, context for this experiment appears in Figure S7. The authors should introduce Figure S7 earlier and as it is currently not mentioned anywhere in the main text, should be called out in the Material and Methods, probably at Line 71.

Thank you for pointing out this issue. The figure with weather data for field trials during four growing seasons is now introduced in the results text each time when we refer to field trial data.

3. For Figure S6, the order of the panels is not the same as described in the figure legend and the associated text in Lines 145-159. It appears panels a and c need to be swapped.

We apologize for this mistake which has been corrected in the revised version of the manuscript.

4. One question that arises for the RT-PCR assays in Miscanthus, where the studied accessions vary in both ploidy (Msa tetraploid, Mxg triploid, Msi diploid) and possibly alleles, is whether all possible copies of the gene are amplified by the assay primers in all accessions? The description of RT-PCR primer design in lines 460-464 is confusing on this point, where it is stated that degenerate primers were designed to target shared sequences across multiple grass species, yet Table S1 lists only one sequence for each primer without any indication of degenerate bases. Did the authors actually mean the opposite, that primers were designed to target gene sequences conserved across the grasses? If degenerate primers were used, then the positions of the degenerate bases should be indicated in Table S1.

We apologize for the incorrect wording. Primers were designed to target gene sequences conserved across the genomes of grasses. The corresponding text in the Materials has been corrected in the revised version of the manuscript.

Point-by-point response to the referees' comments

Referee's original text in black

Our responses in blue

Referee expertise:

Referee #1: stress physiology and molecular biology of plant under the salt and temperature tolerance.

Referee #2: structure, function and regulation of the photosynthetic apparatus and applied research combining photosynthesis.

Referee #3: functional genomics studies of genes that modulate productivity and sustainability in maize and related crop species: sorghum, Miscanthus, and sugarcane.

Reviewers' comments:

Reviewer #1 (Remarks to the Author):

I am satisfied with the authors' response and the manuscript can be accepted for publication.

Thank you for this decision.

Reviewer #2 (Remarks to the Author):

The manuscript has been revised according to the reviewers' comments, has been significantly improved and is now a nice, compact story. The manuscript presents a new phenomenon that is physiologically interesting and potentially very important for future applications. Overall, the data and the interpretation of the data are rather superficial, but on the other hand the main observation is important.

Here are some minor comments.

Line 20: I would soften the statement in the last sentence of the abstract. There is always a trade-off between protection and efficiency.

The new sentence has been added to the abstract to acknowledge the fact that there is a trade-off between photoprotection and the efficiency of photosynthesis.

L24-26: "Driving the engineered trait by chilling inducible promoter would ensure the minimization of a trade-off between photoprotection and photosynthesis efficiency."

Line 34: It is still debatable how sensitive NPQ mutants (PSBS and xanthophyll cycle) are and how well reaction centre quenching can compensate for the lack of these classical NPQ mechanisms.

We agree that "indispensable" might be a too strong word in light of the literature body on NPQ mutants showing varied levels of sensitivity to the light and environmental perturbations. Therefore, we deleted "indispensable".

L38-39: "NPQ is rapid protection from photodamage⁹ that contributes to defense against stress^{10,11}."

Results and discussion: Traditionally, NPQ mechanisms have been seen as protective mechanisms for light reactions, but as this manuscript and some other articles suggest, NPQ may be more important in maintaining source-sink balance, preventing adverse stromal responses and being involved in the control of many physiological responses in which chloroplasts are involved. It would be good to have a little more discussion about these questions whether NPQ is a photoprotective mechanism of light responses or whether it has a more important role in source-sink regulation and in chloroplast-related physiology.

The new text has been added to the Results and Discussion section indicating that NPQ mechanisms might contribute in high chilling tolerant accession of Miscanthus to the maintaining source-sink balance with the addition of appropriate citation.

L108-110: "Upregulation of NPQ via chilling could also contribute to maintaining a balance between source (production of photosynthesis) and sink (consumption of reduced carbon in plant growth)²⁹"

Both redox reagents and nigericin were used. However, the redox regulation of ATP synthase was not discussed. How NPQ is induced upon light onset is strongly dependent on the state of ATP synthase and this should be taken in account in the interpretation of results and in discussion.

We apologize for the overlook of the DTT effect on ATP in our interpretation and discussion of data. The new text has been added to the Results and Discussion section with the addition of appropriate citations.

L151-155: "While ATPase also possesses the thioredoxin-mediated redox modulation, it cannot be ruled out that the observed higher NPQ in MsaRB after infiltration with a strong reductant is due to chloroplast ATPase being more sensitive to reducing environment which would decline its activity leading to higher light-driven proton motive force, which permits for an increase of NPQ^{35,36}."

Reviewer #3 (Remarks to the Author):

The revised manuscript is much improved in its clarity, particularly Figure 1. The authors satisfactorily addressed the comments raised in my first review. I do offer the following suggested minor edits:

1. The proposed change to the abstract to soften the conclusion of "mechanisms" for regulation of NPQ are beneficial, but perhaps could be worded more clearly. I suggest, "Further studies obtained evidence for post-translational modifications to VDE, VDE cofactor accessibility, and lack of upregulation of zeaxanthin epoxidase as possibly contributing to this unique regulation of NPQ."

The text has been changed according to suggestion.

L18-22: "Based on our data, the emerging ways for the unique regulation of NPQ include post-translational regulation of violaxanthin de-epoxidase (VDE), VDE cofactor accessibility, and absence of transcriptional upregulation of zeaxanthin conversion back to violaxanthin. In the future, more studies will be required to obtain further evidence for these ways contributions to the chilling-dark regulation of NPQ."

2. The new text in first paragraph of Results should revise verb tenses, "use" should be "used", "long-living" should be "long-lived" or "prolonged quenching induced by stress".

The sentence has been rewritten.

L72-74: "Under stress conditions the maximal fluorescence (F_m) of dark-adapted leaf used in the formula for $F_v/F_m [(F_m - F_o)/F_m]$; where F_o is the minimal fluorescence] might be underestimated due to stress-related long-lived quenching^{20,22-26}"

3. Bottom paragraph of page 4 Results: "Transcript of VDE transcript levels", "VDE transcript levels"

Thank you for finding this mistake. The sentence has been corrected.